# Effects of Rehabilitation Models on Self-Stigma among Persons with Mental Illness

**DOI:** 10.3390/healthcare10020213

**Published:** 2022-01-22

**Authors:** Yao-Yu Lin, Mei-Ling Lin, Yao-Hui Huang, Wei-Fen Ma, Wen-Jiuan Yen, Shih-Kai Lee

**Affiliations:** 1Tsaotun Psychiatric Center, Ministry of Health and Welfare, Nan-Tou 54249, Taiwan; yylin@ttpc.mohw.gov.tw (Y.-Y.L.); yhhuang@ttpc.mohw.gov.tw (Y.-H.H.); sklee@ttpc.mohw.gov.tw (S.-K.L.); 2College of Nursing, Huangkung University, Taichung 433304, Taiwan; linml@sunrise.hk.edu.tw; 3School of Nursing, China Medical University, Taichung 406040, Taiwan; lhdaisy@mail.cmu.edu.tw; 4Department of Nursing, Chung Shan Medical University, Taichung 40201, Taiwan; 5Chung Shan Medical University Hospital, Taichung 40201, Taiwan

**Keywords:** rehabilitation, self-stigma, mentally ill, generalized estimation equation (GEE), stigma, community, institution

## Abstract

Social stigma is inevitable for mentally ill patients, but how patients treat themselves is a priority for rehabilitation and an important buffer mechanism. This study thus aimed to measure the effectiveness of rehabilitation models for improving self-stigma. This quasi-experimental research design applied purposeful sampling. The participants (*n* = 250) were persons with mental illness who received rehabilitation treatment in central Taiwan. They were divided into community- (*n* = 170) and institution-based (*n* = 80) rehabilitation groups. The Internalized Stigma of Mental Illness Scale was evaluated at the time of recruitment, and a follow-up was conducted after 1 and 3 months. A generalized estimation equation was used in data analysis to measure whether self-stigma improved with the rehabilitation model and time, and to test the effect of different rehabilitation models on participants’ self-stigma improvement. The study found that the self-stigma of patients receiving CBR improved more than that of those receiving IBR when behavioral problems, education, OT level, sex, and first-time self-stigma were controlled. Returning to the community is the goal of rehabilitation for patients with mental illness, but IBR still dominates the rehabilitation model in Taiwan. Thus, it is necessary to continue promoting CBR plans for future mental health policies.

## 1. Introduction

The proportion of persons with chronic mental illness has increased annually in Taiwan. Even though there is more knowledge now regarding mental illness, the public still stigmatizes patients affected by it [1]. Most people’s perceptions of persons with mental illness are that they are violent, bizarre, or disabled [2], such that the public avoids persons with mental illness [3]. What is more, these prejudices influence patients’ adaptation to the community [4].

Driven by the deinstitutionalization policy, the social adaptation of persons with mental illness is an important issue because, even if treatment is successful, the patient’s progress may be affected by returning to a hostile community [5]. The public’s attitude toward persons with mental illness has a negative impact on their work and may devalue them. Furthermore, patients with mental illness may experience lowered self-esteem, self-efficacy, righteous anger, or indifference [6]. Research on the stigmatization of persons with mental illness has a long history. Starting from the earliest research on social stigma, the research focus has gradually shifted to the self-stigma of patients [7]. Self-stigma occurs when a person internalizes the stigma endorsed by the society [8]. A previous study showed that an increase in self-stigma reduced recovery after 1 and 2 years. Therefore, reducing self-stigma is an urgent intervention to improve recovery [9]. Most persons with mental illness participate in rehabilitation programs that include community-based rehabilitation models (CBRs) and institution-based rehabilitation models (IBRs). This study explored the benefits of the two rehabilitation models in reducing self-stigma, which can provide an empirical basis for promoting appropriate rehabilitation programs for patients with mental illness in the future.

### Background

Deinstitutionalization has shifted persons with mental illness from hospitals to the community [10,11,12]. The concept of recovery from mental illnesses is ambiguous. In terms of health policies and services, the focus is on symptom relief and treatment compliance. From an individual’s point of view, it is community involvement, tolerance, a sense of belonging, and happiness [13,14,15].

The recovery issue of persons with mental illness has attracted the attention of scholars. Most rehabilitation studies used qualitative studies, including the subjective experiences of patients [16,17], or quantitative studies to explore patients’ social function, disability level, intellectual function, and cognitive function. Mental function, intellectual function, cognitive function, and family function are also used as indicators of recovery [18,19,20]. In addition, a study also explored the impact of disease management programs on the rehabilitation of mental patients after intervention [21].

Scholars proposed that the goal of rehabilitation for persons with mental illness returning to the community is not only to control symptoms, but also to focus on overall health and self-directed rehabilitation, such as living a satisfactory and autonomous life in the community [22]. Social prejudice is a negative stigma on persons with mental illness, which is a barrier to patients’ adaptation to the community [4].

Stigma is a mark that exists in human society and represents discrimination and prejudice [23]. The stigma regarding mental illness persists and has a wide range of harmful effects on recovery [24]. Stigma may occur in a patient’s sense of self. An individual who agrees with the public’s stigmatized attitude toward themselves [25] may internalize social stigma, thereby exacerbating harm to themselves [6,26] and lowering self-esteem and self-efficacy [27,28]. Patients internalize social prejudice and beliefs about the public’s attitudes toward mental illness, which is called self-stigma. Conversely, stigma resistance is defined as a person’s ability to deflect or challenge stigma beliefs [29]. If the person disagrees with these prejudices, their self-esteem will be greater, and they will feel more empowered [30,31]. Self-stigma is a transformation process in which a person’s social identity is gradually replaced by an “ill identity” with devaluing and stigma characteristics [7]. Scholars have proposed that self-stigma and empowerment are two extreme linear relationships that represent negative attitudes toward illness, leading to low self-esteem; conversely, self-empowerment represents positive self-esteem [32,33]. Schmader, Major, and Gramzow [33] explain that the excitement associated with empowerment is a defense against social stigma. Interventions to reduce self-stigma may benefit patient recovery [34]. If a person with mental illness has a negative self-concept, they will hide their mental illness and refuse treatment [35]. Hence, self-stigma is an important psychological rehabilitation mechanism [31], and it affects recovery [27].

Taiwan’s mental health care model changed from institutionalized to patient-needs-centered community rehabilitation in 1989 [36]. Assisting patients to successfully adapt to the community has become a new trend in care, including daycare centers and residential mental health rehabilitation institutions (half-way houses). Overall, rehabilitation centers can be classified into community-based rehabilitation (CBR) and institution-based rehabilitation (IBR) [37]. The ideal and most familiar environment for patients is the community in which they live. Therefore, community rehabilitation models have become the mainstream treatment for persons with chronic mental illnesses in various countries [38].

Regardless of the models, the purpose of rehabilitation is to build a patient’s ability to work, work attitude, social skills, and daily life processing skills so that they can live independently and improve their functions to adapt to family and social life [16,17]. Studies have found that the rate of readmission varies according to different rehabilitation models [16,39]. Patients receive continuous follow-up in IBR, but CBR allows patients to attain more opportunities to become involved in society. There is no consensus as to which model is better; therefore, the benefits of different rehabilitation models are worth exploring. Based on the above review, in this study, we are interested in understanding the differences in patients’ self-stigma based on the different rehabilitation models they receive. The research results can be used as an empirical basis for the development of future rehabilitation models.

## 2. Materials and Methods

### 2.1. Participants

This was a quasi-experimental research design, and purposive sampling was applied. The participants were recruited from a psychiatric center in central Taiwan. The persons were discharged from the psychiatric centers, transferred to the daycare center, and then assigned to the IBR group, and those transferred to the affiliated community rehabilitation center were assigned to the CBR group.

Based on the International Disease Classification (ICD-10) system used by the Health Insurance Agency, the inclusion criteria were the diagnosis of a mental disorder (IDC 295) or an emotional psychiatric disorder (IDC 296) in those aged 20–65 years currently receiving a rehabilitation program, and positive or negative symptoms not affecting data collection, communicability, or willingness to participate in this research.

Exclusion criteria included people who were at obvious risk of violence or suicide, those with a dual mental illness diagnosis (for example, substance abuse or personality disorder), those who were suspected of having insufficient intelligence, or those who had been diagnosed with serious medical problems and needed treatment.

Considering the patient’s adaptation to the rehabilitation program, each participant was contacted by a research team member 1 week after being admitted to the rehabilitation program. The purpose of the study was explained first, and after obtaining the participants’ consent, the participants were recruited in the study.

### 2.2. Sample Estimate

Our study measured the difference in self-stigma between the two models over time; therefore, we estimated the sample using G* power with repeated measures MANOVA, alpha was set to 0.05, power was set to 0.85, and the medium effect size was set to 0.25, with two groups and three measurements. The estimation required 178 participants, with approximately 90 participants in each group. To avoid invalid questionnaires, we recruited 125 participants from each group, and a total of 250 participants were recruited.

### 2.3. Instruments

The Internalized Stigma of Mental Illness Scale (ISMIS) is a 29-item, 4-point Likert-type scale with five constructs: (1) alienation: refers to the sense of self about their integration into society; (2) stereotype endorsement: personal stereotypes toward people with mental illness; (3) discrimination experience: this level of individual prejudice against persons with mental illness; (4) social withdrawal, social withdrawal resulting from social stigma; and (5) stigma resistance (reverse-coded): patients’ coping with stigma. A higher score represents more serious self-stigma. The reliability of this scale was tested for internal consistency and retest reliability, and its validity was tested using factor analysis for convergent validity [40]. The current reliability was measured by internal consistency, with a Cronbach’s alpha of 0.94.

### 2.4. Procedure

Participants were contacted and the purpose of the study was explained to them. After their consent was obtained, the first questionnaire interview and evaluation were started as the benchmark, and then data were collected at the first and third month marks from the initial interview and evaluation. The study was approved by the institutional review board for the protection of human participants’ rights at one hospital.

### 2.5. Data Analysis

Data were analyzed using SPSS/PC for the Windows. Frequency, percentage, average, and standard deviation were used to analyze descriptive data. Inferential analysis of the chi-squared tests and independent *t*-tests were used to compare participants’ characteristics at baseline. The generalized estimating equation (GEE) model was used to examine the effects of the rehabilitation program. To reduce the possibility of committing a type II error, variables with a *p*-value of <0.05 in the baseline comparisons between groups were treated as covariates in the adjusted GEE models.

### 2.6. Ethical Considerations

The study was submitted to the Institutional Review Board for the protection of human subjects at a psychiatric center in central Taiwan for approval human rights (IRB No. 104045). Patients were recruited after obtaining their permission. Data were collected in a private hospital setting and conducted by the first author in Mandarin or Taiwanese, which are the main local languages in Taiwan.

## 3. Results

### 3.1. Demographic Data

A total of 250 participants were surveyed—170 from the CBR and 80 from the IBR (Table 1)—and a *t*-test was conducted to compare the two groups. Of the 250 participants, 133 were men (53.2%) and 117 were women (46.8%). A total of 79.2% (*n* = 198) of the sample were married or had partners and were thus classified into the couple group. Single, divorced, or widowed persons were classified into the no-couple group. The educational level was mostly junior high or below (*n* = 131, 52.4%). The main disease diagnosis was schizophrenia (*n* = 178, 71.2%), and 40.4% (*n* = 101) of the participants were classified as having mid–low occupational therapy (OT) (levels (52.4%). The average age of the participants was approximately 47.96 (*SD* = 11.12), and the average number of years of illness was 20.08 (*SD* = 10.11). The mean level of behavioral problems was 9.23 (*SD* = 3.26), and a significant difference (*t* = 2.96, *p* < 0.001) was found between CBR (9.63, *SD* = 3.28) and IBR (8.38, *SD* = 3.07). The average self-stigma was 68.16 (11.56), and a significant difference was found between the two groups. That is, the self-stigma of CBR (71.22, *SD* = 10.62) was higher than that of IBR (61.66, *SD* = 10.85). Additionally, sex, couple, education, OT level, behavioral problems, and first-time self-stigma (SS1) were found to be significantly different between CBR and IBR, so these variables were identified as covariants in the GEE test.

### 3.2. The Effect of Self-Stigma Reduction with Different Rehabilitation Models

Table 2 and Figure 1 both indicate that the self-stigma of the CBR group was higher than that of the IBR group. Based on this research question, the researchers used a GEE to estimate the parameters of a generalized linear model with a possible correlation between outcomes. Before data analysis, a normal distribution was performed. The ratio of the mean to the standard deviation, the mean and the median, and skewness and kurtosis were examined, and the results show that the mean was almost equal to the median. Skewness and kurtosis were almost between −1 and 1; therefore, the data conform to the normal distribution hypothesis. Baseline comparisons between groups revealed that sex, couple, education, OT level, behavior disorder, and self-stigma pre-test were significantly different, and these variables were treated as covariates.

The GEE results (Table 3) showed that time and rehabilitation models had no direct effect but showed an interaction; that is, as time passed, the self-stigma of patients receiving community rehabilitation gradually decreased (B= −1.72, X^2^ = 5.65, *p* < 0.05). The interaction between CBR and time was significant only when comparing Time 3 to Time 1 when the variables were controlled, including behavioral problems, education, OT level, sex, and first-time self-stigma were controlled.

## 4. Discussion

The average age of the patients was 46–47 years, and the number of ill years was approximately 19–20 years in this study. If these cases cannot return to the community, we can expect that long-term care of mentally ill patients will become a national concern in the future. Therefore, the rehabilitation of persons with mental illness is a very important issue.

We considered rehabilitation of persons with mental illness to be the best treatment for returning to the community. However, social attitudes toward mental illness affect the adjustment of persons with mental illness in society; a positive attitude makes patients feel supported and included. Social attitudes affect a patient’s self-concept. If the patient internalizes a negative attitude, it may result in self-stigma. In this study, first-time self-stigma (Table 1) was measured 1 week after joining the rehabilitation program. We used a *t*-test to examine the first-time self-stigma difference between CBR and IBR, which showed that the self-stigma of CBR was higher than that of IBR. Patients in the community may face many practical difficulties, coupled with public prejudice, increasing their sense of self-stigma. Compared with IBR patients, they are less exposed to community pressure or complicated interpersonal communication, which can consciously reduce self-stigma [41].

When we control for covariates, including behavioral problems, education, OT level, sex, and first self-stigma, there is an interaction between rehabilitation models and time. Initially, patients receiving CBR had a higher level of self-stigma than patients receiving IBR; however, over time, patients receiving CBR had less self-stigma than patients receiving IBR. Best [42] explained that positive social recognition and active community engagement can change internalized stigma. Zerdila et al. [43] investigated the satisfaction of mentally ill patients living in community-based settings, and found that most people felt safe and more satisfied because they could contact and communicate with relatives and friends. The social environment, especially close relationships, has a very important impact on mental health [44]. CBR provides a real environment for patients to interact with and become involved in. CBR participants can get rid of the stereotype toward the hospital and easily come into contact with the general public, meaning that they have a better sense of social integration. A sense of belonging and hope are both important factors for patient recovery and are essential to the human experience [45]. Staying in the community can maintain interpersonal relationships and employment, thereby eliminating negative symptoms and improving life satisfaction [46,47]. McInerney et al. [47] examined the quality of life and social function of long-term mentally ill inpatients after being transferred to the community. They found that even among those who had been in the institution for many years, when they were transferred into the community, their quality of life and social function also improved. Although the above reference showed that patients’ social adaptation in CBR is better than IBR, it is undeniable that IBR provides a protective environment for patients. Patients receive respect and professional care and are unlikely to encounter real difficulties. [41]. This explains why the self-stigma of IBR was lower than that of CBR when the patients were first recruited.

Self-stigma refers to how the patient thinks about themselves, similar to cognitive theory in psychology. The main concept of cognitive theory lies in the individual’s interpretation of the event, rather than the event itself. Similarly, we do not focus on social prejudice, but on how patients perceive themselves. This is a very important driver for maintaining a positive attitude toward life. There are few studies in Taiwan that use self-stigma as a rehabilitation indicator. The results of this study provide good evidence for the future development of community rehabilitation models. The results of this study support the notion that the self-stigma improvement of CBR is better than that of IBR, but IBR still dominates the rehabilitation model in Taiwan. In any case, patients must return to the community. CBR is a care model that must be developed in the future; however, the prejudice of the public and the lack of community care experts make it difficult to expand the function of community rehabilitation models.

Based on the results, nursing staff must actively extend their expertise from the hospital to the community and improve community nursing knowledge and skills, including assessment, referral mechanisms, and interventions. In terms of education, it is necessary to strengthen the concept of community care for mentally ill patients and guide students to think about the dilemmas that patients may face in the community. Only through integration of policy, practice, and education to establish a supportive community mental health model can patients return to the community as early as possible. This research can be used as empirical support material to promote community mental health care.

The inferences of this study need to consider the following limitations. As far as the research objects are concerned, the participants’ diagnosis was mainly schizophrenia, so it is impossible to estimate patients with other mental illness diagnoses or with different characteristics. In terms of research variables, since it does not include the patient’s self-esteem and previous experience in receiving psychological rehabilitation services, it is also impossible to clearly determine the impact of these variables on effectiveness. This was a quasi-experimental study, and its causal inferences were not conclusive despite of its covariate analysis.

## 5. Conclusions

Following deinstitutionalization, patients with mental illness returning to the community become the treatment target. Medications have limited effects on improving patients’ negative symptoms, cognitive deficits, self-care, social skills, interpersonal relationships, employment, and leisure activities. Psychological rehabilitation is needed to restore patients’ physical and mental health and for them to return to society [34]. With the deinstitutionalization movement, the Taiwanese government has paid increasing attention to community rehabilitation, but in terms of proportion, only a limited number of patients use CBR [5]. Wang and Ouyang [34] analyzed the problems of community rehabilitation in Taiwan, including uneven resource allocation, an insufficient number of community rehabilitation centers, insufficient national medical insurance payments, poor community service quality, and lack of continuous community care in case management plans. Furthermore, excessive emphasis on vocational training, the ignorance of psychological rehabilitation, and the failure to give family members and patients appropriate empowerment in the process of psychological rehabilitation results in patients failing to actively participate in community rehabilitation. Our research results provide strong evidence that the self-stigma of patients receiving CBR is lower than that of those receiving IBR. We believe that these results can be used as a reference to promote future community rehabilitation programs.

## Figures and Tables

**Figure 1 healthcare-10-00213-f001:**
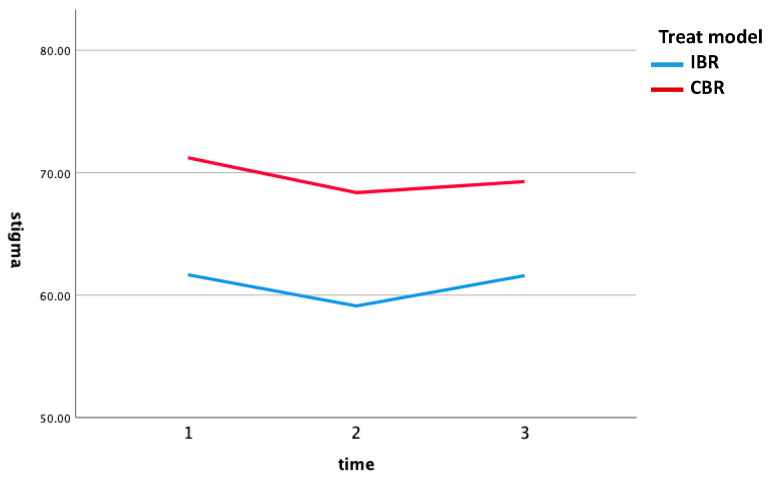
Self-stigma*time of CBR and IBR groups.

**Table 1 healthcare-10-00213-t001:** Demographic difference between CBR and IBR groups. *N* = 250.

Variable	Total	CBR (*n* =170)	IBR (*n* = 80)	t/x^2^
	*n* (m/sd)%	*n* (m/sd)%	*n* (m/sd)%	
Sex				6.58 **
Men	133 53.2	81 47.65	52 65	
Women	117 46.8	89 52.35	28 35	
Couple				20.76 ***
Couple	198 79.2	121 71.18	77 96.25	
No-couple	52 20.8	49 28.82	3 3.75	
Education				15.77 **
Junior high or below	131 52.4	104 61.18	28 35	
Above high school	118 47.6	66 38.82	52 65	
OT level				21.66 ***
Mid-low	170 52.4	96 56.47	20 25	
High	80 47.6	74 43.53	60 75	
Diagnosis				2.09
Schizo	178 71.2	116 68.24	62 77.5	
Non-schizo	71 28.4	54 31.76	18 22.5	
Age	47.96 (11.12)	48.57 (11.79)	46.65 (9.45)	1.38
Ill years	20.08 (10.11)	20.21 (10.04)	19.80 (9.38)	0.30
Behavior	9.23 (3.26)	9.63 (3.28)	8.38 (3.07)	2.96 **
Self stigma	68.16 (11.56)	71.22 (10.62)	61.66 (10.82)	6.55 ***

CBR, community-based rehabilitation. IBR, institution-based rehabilitation. ** *p* < 0.001, *** *p* < 0.001.

**Table 2 healthcare-10-00213-t002:** Self-stigma between CBR and IBR groups.

		N	Range	Mean	SD	95% CI	
						Lower	Upper
SS1	IBR	80	29.00~98.00	61.6625	10.82887	59.2527	64.0723
	CBR	170	45.00~112.00	71.2176	10.61732	69.6101	72.8252
SS2	IBR	78	28.00~94.00	59.1154	10.36976	56.7774	61.4534
	CBR	170	43.00~109.00	68.3706	10.31505	66.8088	69.9324
SS3	IBR	80	29.00~98.00	61.6000	11.32780	59.0791	64.1209
	CBR	169	45.00~112.00	70.8284	10.39460	69.2499	72.4069

SS1, first-time self-stigma; SS2, second-time self-stigma; SS3, third-time self-stigma.

**Table 3 healthcare-10-00213-t003:** Effects of the rehabilitation model on self-stigma.

Variables	B	SE	95% CI		Wald X^2^	*p*-Value
			Lower	Upper		
(intercept)	5.143	2.5190	0.206	10.080	4.169	0.041 *
CBR	−1.330	3.2894	−7.778	5.117	0.164	0.686
IBR	^a^					
[time = 3]	−0.230	0.6425	−1.489	1.029	0.128	0.721
[time = 2]	−2.524	0.7122	−3.920	−1.128	12.553	0.000 ***
[time = 1]						
Above high school	−1.664	0.6671	−2.972	−0.357	6.223	0.013 **
Junior high or below						
Couple	−0.013	0.8969	−1.771	1.745	0.000	0.988
No-couple	^a^					
High OT level	−0.559	0.6793	−1.891	0.772	0.678	0.410
Mid-low OT level	0 ^a^	.	.	.	.	.
Men	−0.331	0.6371	−1.579	0.918	0.269	0.604
Women (reference)						
Behavior disorders	0.282	0.1100	0.066	0.497	6.556	0.010 **
Pretest self-stigma	0.907	0.0301	0.848	0.966	910.329	0.000 ***
CBR * [time = 3]	−1.717	0.7227	−3.134	−0.301	5.647	0.017 *
CBR * [time = 2]	−0.0324	0.7780	−1.848	1.201	0.173	0.678
CBR * [time = 1]						
IBR * [time = 3]						
IBR * [time = 2]						
IBR * [time = 1]						
CBR * high OT level	0.678	0.7432	−0.778	2.135	0.833	0.361
CBR * mid-low OT level						
IBR * high OT level						
IBR * mid-low OT level						
CBR * bev01	−0.304	0.1150	−0.530	−0.079	7.003	0.008 **
IBR * bev01		.	.	.	.	.
CBR * first-time self-stigma	0.041	0.0418	−0.041	0.123	0.954	0.329
IBR * first-time self-stigma						

* *p* < 0.05 ** *p* < 0.01. *** *p* < 0.001. ^a^ It is comparison group.

## Data Availability

Raw data were generated at the Tsao Psychiatric Center, Taiwan. The derived data supporting the findings of this study are available from the corresponding author Yen, W.J. on request.

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
