# Peer review of "Effects of Rehabilitation Models on Self-Stigma among Persons with Mental Illness"

_healthcare, 2022, doi:10.3390/healthcare10020213_

Round 1
Reviewer 1 Report
The manuscript reports a study on the comparison between two rehabilitation models in terms of their effects on self-stigma. The topic is important, and the three-wave research is notable in terms of the effort involved on such a specific and less accessible sample. There are nevertheless several issues that need to be addressed in order to improve the manuscript.
- Abstract: “The purpose of the study was to measure the effectiveness of a rehabilitation model for improving self-stigma” – should be “two models”, as both were tested
- CBR and IBR should be defined in the Abstract the first time they are used;
- line 23 “self-stigma, and interaction with rehabilitation models” – should be “interacts”
- line 112 “exacerbating harm to themselves [6,19], self-esteem, and self-efficacy [20,21].” – I suppose it should be “lowering self-esteem”
- the last two paragraphs on page 2 in the Introduction should be reorganized: the definition of self-stigma should be placed before its effects (now described in the first paragraph). Also, the results of resisting stigma, now included in both paragraphs, should be merged;
- The Introduction should include a review of past research on the effects of the two rehabilitation models, on the different facets of mental and psycho-social rehabilitation that have been examined.
- there should be an explanation / argument regarding the authors’ decision to set the MANOVA effect size to 0.25.
- Table 1: the description of the data in the text does not fully correspond to the data in the table: what is high-way houses? Also, the variable number of years of illness is not presented in the table; OT level should be explained in the Notes; the phrase “and 40.4% (n = 101) participants were receiving high-level rehabilitation (53.6%) and came from half-way houses.” needs correction in the percentages reported (there are two percentages for the same category); the SDs of the quantitative variables are not reported in the table. The variable “Behavior” is unclear, and it was not referred to before in the Methods. “Behavior disorders” appear in Table 3, but there’s no detail about how they were measured. All these issued should be fixed.
- Section 2.1. states that “Participants were recruited from a psychiatric center in central Taiwan, including outpatients and affiliated community rehabilitation centers”. Does this imply that some participants were at the beginning of the study hospitalized in the psychiatric center? Are these the participants included in the IBR condition in the study design? And the others are “outpatients and affiliated community rehabilitation centers”? Are these the participants in the CBR condition? These aspects are unclear. Furthermore, there is no data concerning the time since participants were included in the two rehabilitation models, at least in the CBR (I suppose that the IBR might mean the actual treatment in the psychiatric center, but I might be wrong). Moreover, this variable (the time since the admission in the respective rehabilitation model) could be quite influential on stigma and self-stigma, therefore it should be controlled when assessing the differences between the two models. Alternatively, it should be at least acknowledged as a major limitation of the study.
- Table 2 – what are SS1 to SS3? They should be defined in the Notes.
- Results: the manuscript states (line 229) that time has no effect, but time=2 is significant (thus there is a significant variation in stigma from the first to the second time of measurement).
- Statistic indicators should not appear both in the text and in the Table (i.e., Table 3).
- line 230 “as time passed, the self-stigma of patients receiving the community rehabilitation gradually decreased” – the interaction CBR x time was significant only when comparing time 3 to time 1, this should be mentioned.
- line 231 “Behavioral problems were found to have a direct effect on self-stigma (X2=6.56, p<.01) and interaction with rehabilitation models (B = -.30, X2 = 7.00, p < .01). This means that when the variable, behavioral disorder, is controlled, people who accept CBR will have lower self-stigma than IBR.” The interpretation of the interaction is not adequate: behavioral disorder is controlled by its inclusion in the set of predictors / independent variables; thus, the effect of CBR (when compared to IBR) when controlling behavioral problems is shown on the second line of Table 3, as is not significant (p = .686). The significant interaction implies that the effect of CBR is different in one of the categories / at one end of the measure of behavioral problems when compared to the other(s) (since there no info on measuring this variable, it’s unclear how this interaction might in fact entail). Thus, the data analysis should be extended by examining the effect of CBR vs. IBR in each category of behavioral problems on stigma (could be on the overall mean of the three measures of stigma). Relatedly, line 260 in the Discussion states “Patients with more behavioral problems, after receiving CBR, showed, lesser self-stigma than those receiving IBR”, but there’s no support for this statement included in the Results.
- Table 3 is not properly labeled, it should announce the effect presented there (of which variables on which variable), and not the data analysis technique (i.e., GEE outcome);
- Discussion, line 244: “patients who received CBR had a higher level of self-stigma than those receiving IBR” – I disagree, the difference was not significant.
- line 280 I don’t see the connection between the results of Lien and Kao [49] and the finding that “finding that schizophrenic patients with higher education levels and pre-stigma demonstrate higher post-self stigma.”
- the benefits of CBR are reviewed and discussed in this section, but there should be a parallel and complementary discussion on the factors that decrease / hinder the rehabilitation benefits of IBR; as the manuscript states (line 278) “Mental illness collapses the self-esteem of highly educated patients and leads to higher self-stigma [41].” – why? by which mechanisms of psychological or social influence? This would be important in order to allow the reader to appreciate the determinants of the differences that the study found between the two models. Specifically, are they due to CBR offering benefits of social interaction, awareness, etc.? Or are they determined by merely cutting the exposure of the mental patient to the negative factors involved by the IBR?
Most of the phrases in the Conclusion should be integrated in the Discussion; the conclusions should only refer to the results of the study and briefly state their implications.
Author Response
Thank you very much for reviewing my manuscript and I try my best to revise it.
1. We modified table1 to match the results
2. For GEE results, behavioral problems are considered as covariates and should be controlled. Our previous explanation was wrong and has been corrected.
3. In the discussion section, we added some suggestions for clinical practice and education.
All corrections are written in red.
Reviewer 2 Report
In this article, the authors compared the effects of two rehabilitation approaches, namely the community-based rehabilitation model (CBR) and the institution-based rehabilitation model (IBR) in reducing self-stigma in psychiatric patients. Interestingly, the self-stigma of patients receiving CBR improves more than that of those receiving IBR. Moreover, education and pre-testing self-stigma have a direct impact on self-stigma. The higher the educational degree, the lower the self-stigma. The higher the initial test self-stigma, the higher the post-test self-stigma.
In my opinion, this manuscript is well written. I only have minor comments, which may help improving the manuscript before publication in Healthcare (section Nursing).
1) In the Abstract, CBR and IBR acronyms appear in the text for the first time, but they are not spelled out.
2) Section 1. Introduction and subsection 1.1 background are very similar and sometimes repetitive. I suggest putting them together in a single paragraph.
3) I suggest carefully reading the manuscript to avoid typos.
4) When describing self-stigma in the background section, can the authors specify whether patients affected by a specific psychiatric disorder are more self-stigmatized than other psychiatric patients are?
5) In Materials and Methods, section 2.1 Participants, the sentence “in those aged 20-65 years currently receiving a rehabilitation program, positive or negative symptoms did not affect data collection, communicability, or willingness to participate in this research.” needs further explanations.
6) In the same paragraph and in Table 1, did the authors take into account the comorbidity with other illnesses, including for instance cardiovascular and metabolic disorders? Did the authors observe gender-related differences?
7) Conclusions about the impact of authors’ main findings on intervention strategies and their possible contribution in clinical settings should be better argued, also mentioning to the implementation of future mental health policies.
Author Response
Thank you very much for reviewing my manuscript and I will try my best to revise it.
1. All corrections are written in red
2. There is no information to indicate which diagnosis experience more self-stigma, but in this study, we measured the difference in self-stigma between schizophrenia and non-schizophrenia, and the results showed no difference.
Reviewer 3 Report
A worthwhile research project. The concept of 'self stigmatization' is well explained and supported by referenced research.
The structure of the paper follows a conventional presentation. I have identified quite a number of typographical/grammar errors. The majority being at: 42/43 ?work. para 109 - 117 too many references. 126 Gramzow. 137/138. 151/155 long sentence. 179 not plural. 194. 202 remove 'of them'. 203 remove of the dise3ase. 213.216.219.234 'less'. 216 and overuse. 235. 253 suitable. 254 'be'. 259. 285 with Schizophrenia. 289 's' redundant 290 'the' redundant. Refs required at 34.48/49.
The tables and quantitative analysis appear clear and sound.
Author Response
Thank you very much for reviewing my manuscript and I tried my best to revise it.
1. All corrections are written in red
2. Due to time limit for revicion , we will request English editing of this version
Round 2
Reviewer 1 Report
The manuscript is improved, but some of my previous comments were left unadressed. Moreover, it's hard to follow the modifications that the authors did without a proper Letter of changes that describes the changes made in relation to each comment of the reviewer.
My comments regarding this version are:
From the previous review form:
Abstract: “The purpose of the study was to measure the effectiveness of a rehabilitation model for improving self-stigma” – should be “two models”, as both were tested
- The Introduction should include a review of past research on the effects of the two rehabilitation models, on the different facets of mental and psycho-social rehabilitation that have been examined. In the Discussion, from line 258, the manuscript states “Most rehabilitation research used qualitative 258 study, including subjective experience of patients [37,38], or quantitative study to explore pa-259 tients' social function, disability level, intellectual function, and cognitive function” – such studies should be reviewed in the Introduction.
- the last two paragraphs on page 2 in the Introduction should be reorganized: the definition of self-stigma should be placed before its effects (now described in the first paragraph). Also, the results of past research on the effects of resisting stigma (e.g., disagreeing with these prejudices), now included in both paragraphs, should be merged.
New comments:
- line 153 the phrase “Data collection on IBR patients in daycare centers and recruit CBR patients from the outpatient department.” should be corrected, doesn’t make sense now.
- line 150 “The patient was discharged from the psychiatric cneter” – should be “patients”, and “center”
- line 152 “transferred to the affiliated community rehabilitation center and halfway house to the community rehabilitation group.” – what does “halfway house” mean? and generally, the phrases added here should be re-checked and clarified, this essential part of study sampling it’s hard to understand now.
- Results: the manuscript states (line 229) that time has no effect, but time=2 is significant (thus there is a significant variation in stigma from the first to the second time of measurement).
- Statistic indicators should not appear both in the text and in the Table (i.e., Table 3).
My previous comment relevant here is still relevant: Furthermore, there is no data concerning the specific time since participants were included in the two rehabilitation models. Moreover, this variable (the time since the admission in the respective rehabilitation model) could be quite influential on stigma and self-stigma, therefore it should be controlled when assessing the differences between the two models. Alternatively, it should be at least acknowledged as a major limitation of the study.
Extending another previous comment, there are no limits of the study acknowledged (should be placed at the end of Discussions).
Author Response
Dear reviewer
Thank you very much for your kind and careful review and comments. We modify as follows based on your comments.
